# Revealing carbon capture chemistry with 17-oxygen NMR spectroscopy

Astrid H. Berge [1,6], Suzi M. Pugh[1,6], Marion I. M. Short[1], Chanjot Kaur[2], Ziheng Lu[3], Jung-Hoon Lee[4], Chris J. Pickard [3,5], Abdelhamid Sayari [2] & Alexander C. Forse [1]

Carbon dioxide capture is essential to achieve net-zero emissions. A hurdle to the design of improved capture materials is the lack of adequate tools to characterise how $CO_2$ adsorbs. Solid-state nuclear magnetic resonance (NMR) spectroscopy is a promising probe of $CO_2$ capture, but it remains challenging to distinguish different adsorption products. Here we perform a comprehensive computational investigation of 22 amine-functionalised metal-organic frameworks and discover that $^{17}O$ NMR is a powerful probe of $CO_2$ capture chemistry that provides excellent differentiation of ammonium carbamate and carbamic acid species. The computational findings are supported by $^{17}O$ NMR experiments on a series of $CO_2$-loaded frameworks that clearly identify ammonium carbamate chain formation and provide evidence for a mixed carbamic acid – ammonium carbamate adsorption mode. We further find that carbamic acid formation is more prevalent in this materials class than previously believed. Finally, we show that our methods are readily applicable to other adsorbents, and find support for ammonium carbamate formation in amine-grafted silicas. Our work paves the way for investigations of carbon capture chemistry that can enable materials design.

Carbon dioxide capture and storage is essential to reducing greenhouse gas emissions and meeting net-zero emissions targets[1,2]. A range of technologies are under development to meet the need for more energy efficient carbon capture. One promising strategy to improve on traditional aqueous amine technology is to use solid adsorbent materials for capture[3–5]. In particular, installation of reactive amine or hydroxide functional groups within a porous scaffold such as a metal-organic framework or a porous silica brings about selective reactivity with $CO_2$[6–10], with the porous scaffold providing a large surface area for hosting the reactive groups while maintaining channels for $CO_2$ transport. The increasingly complex adsorbent materials under consideration bring major challenges in the characterisation of new carbon capture chemistry, hindering the design

of improved materials[4]. Existing characterisation tools for understanding $CO_2$ capture modes include single-crystal diffraction[11–13], powder diffraction[14], infra-red spectroscopy[6,10,15], X-ray absorption spectroscopy[16], and NMR spectroscopy[8,17–20], each of which has strengths and limitations in terms of the materials that can be studied and the information that can be obtained. Solid-state NMR spectroscopy is a promising tool for investigating $CO_2$ binding modes in adsorbents as there is no requirement for long-range ordering and detailed information about the local structure and dynamics of the $CO_2$ can be obtained. However, different $CO_2$ adsorption products often give rise to very similar signals in the NMR spectrum and assigning these signals to specific $CO_2$ binding modes remains very challenging[8,17–20].

[1]Department of Chemistry, University of Cambridge, Lensfield Road, Cambridge CB2 1EW, UK. [2]Centre for Catalysis Research and Innovation (CCRI), Department of Chemistry and Biomolecular Sciences, University of Ottawa, Ottawa, Ontario K1N 6N5, Canada. [3]Department of Materials Science and Metallurgy, University of Cambridge, 27 Charles Babbage Road, Cambridge CB3 0FS, UK. [4]Computational Science Research Center, Korea Institute of Science and Technology (KIST), Seoul 02792, Republic of Korea. [5]Advanced Institute for Materials Research, Tohoku University, Aoba, Sendai 980-8577, Japan. [6]These authors contributed equally: Astrid H. Berge, Suzi M. Pugh. ✉e-mail: acf50@cam.ac.uk

The most common experiment with the NMR approach is to dose the candidate adsorbent with $^{13}CO_2$ gas and perform $^{13}C$ magic angle spinning (MAS) NMR experiments. These experiments are relatively straightforward to perform, but often lead to ambiguous identification of the adsorption products. For amine-functionalised materials, the $^{13}C$ chemical shifts give poor differentiation between closely related ammonium carbamate, carbamic acid, and ammonium bicarbonate adsorption products, with the signals from these species showing very similar $^{13}C$ chemical shifts[17,18]. A similar problem arises for bicarbonate and carbonate products in hydroxide-based materials[20]. The prediction of NMR parameters with density-functional theory (DFT) calculations[21] can improve confidence in the structural assignments, and more advanced multi-nuclear NMR experiments can give improved differentiation between adsorption products[17,19,22]. However, there remains a pressing need for the exploration of new NMR methods for understanding $CO_2$ capture chemistry[23].

A representative emerging class of $CO_2$ adsorbents are amine-functionalised metal-organic frameworks. The framework $M_2(dobpdc)$ (dobpdc = 4,4′-dioxidobiphenyl-3,3′-dicarboxylate) (Fig. 1a) can straightforwardly be functionalised with amines to yield a family of (amine)–$M_2(dobpdc)$ adsorbents (Fig. 1b)[14]. These adsorbents have large capacities for selective and reversible $CO_2$ uptake, and the adsorption thermodynamics can be tuned by varying the amine[11,24–27], and the metal[14,28,29]. Importantly, these materials generally display steep adsorption isotherms making them promising for a range of energy efficient carbon capture applications[24,25]. Initial characterisation of $CO_2$ adsorption modes in these materials has revealed a rich chemistry, with three $CO_2$ adsorption products proposed to date: (i) ammonium carbamate chains (Fig. 1c), thought to be the dominant product in a range of variants[11], (ii) carbamic acid pairs (Fig. 1d), identified in the Zn-based framework functionalised with the diamine

dmpn (dmpn = 2,2-dimethyl-1,3-diaminopropane)[17,24], and (iii) a mixed adsorption product (Fig. 1e) recently proposed for (dmpn)-$Mg_2(dobpdc)$[17]. The adsorption thermodynamics of these three adsorption processes vary, motivating further characterisation to aid the design of metal-organic frameworks with the best $CO_2$ capture performances.

Here we leverage the crystalline and tuneable family of (diamine)–$M_2(dobpdc)$ adsorbents to perform a systematic computational exploration of solid-state NMR parameters for different $CO_2$ adsorption products. We show that $^{17}O$ solid-state NMR spectroscopy is a powerful probe of $CO_2$ capture chemistry, providing unambiguous identification of carbamic acid formation and a detailed picture of the hydrogen-bonding environments.

## Results and discussion
### Computational discovery of $^{17}O$ NMR as a probe of carbon capture
The broad tunability of the diamine–$M_2(dobpdc)$ family of materials and large number of available DFT-calculated adsorption structures presented an excellent opportunity to explore NMR parameters for differentiating $CO_2$ adsorption products. To increase the diversity of structures explored, for select materials we explored not only the leading candidate adsorption product from previous studies[11,17], but also the other adsorption products in Fig. 1 (see Supplementary Table 1 for a list of studied materials). While $^{13}C$ NMR spectroscopy is the most commonly used tool for probing $CO_2$ adsorption products, our DFT calculations show that poor differentiation of ammonium carbamate and carbamic acid binding modes is achieved by the $^{13}C$ chemical shifts (Supplementary Fig. 1a, b, Supplementary Table 2). Better differentiation of products is achieved by considering the orientation dependence of the $^{13}C$ chemical shift (i.e., the chemical shift anisotropy),

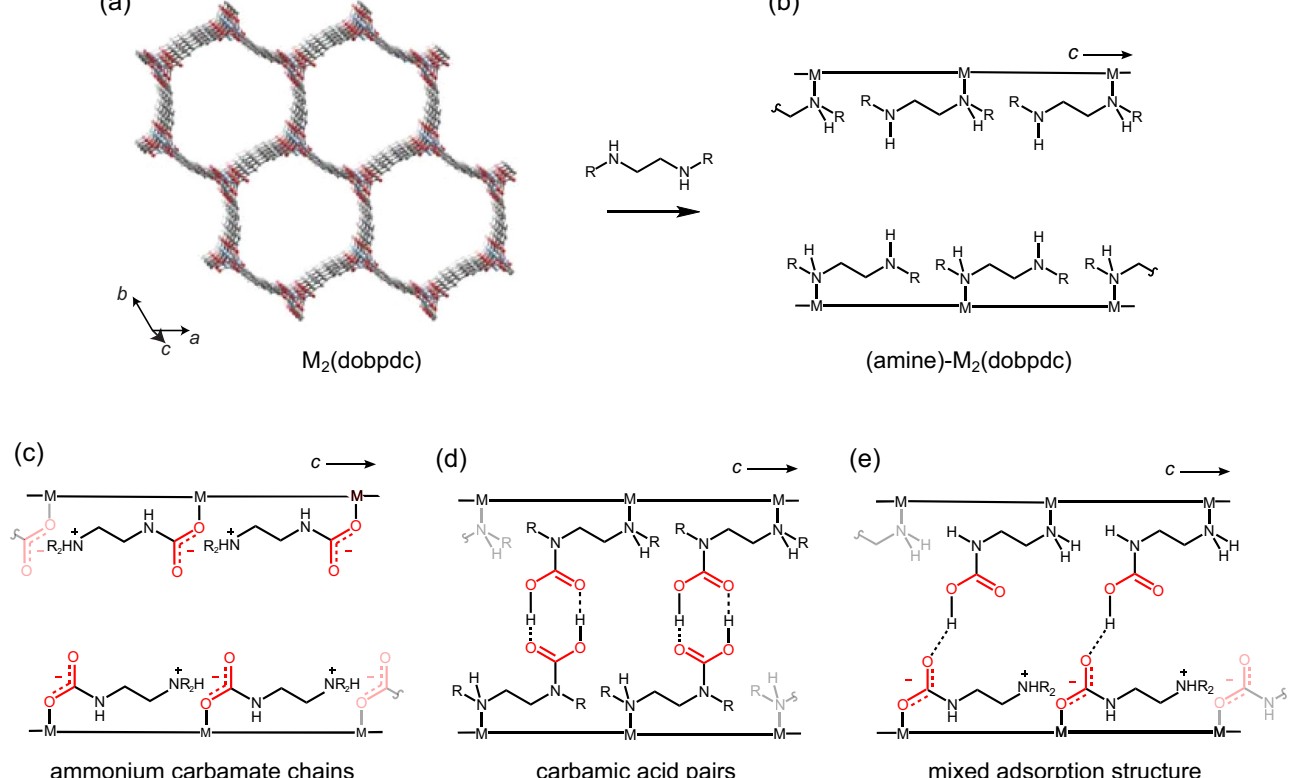

**Fig. 1 | Structure and $CO_2$ adsorption modes in (amine)–$M_2(dobpdc)$ metal-organic frameworks. a** Structure of the metal organic framework $M_2(dobpdc)$. **b** Amine functionalisation yields (amine)-$M_2(dobpdc)$. Upon exposure to $CO_2$ various adsorption products can form. The most dominant adsorption products are: **c** ammonium carbamate chains, **d** carbamic acid pairs and **e** mixed ammonium carbamate–carbamic acid.

consistent with recent work on amine-functionalised silicas[19,23]. However, in many cases carbamic acid and ammonium carbamate species still have similar [13]C NMR parameters (Supplementary Fig. 1c) because both the protonation state (ammonium carbamate chain or carbamic acid) and other hydrogen bonding interactions have an impact. These findings support the idea that alternative NMR probes beyond [13]C are needed to characterise $CO_2$ adsorption modes with greater confidence.

We hypothesised that [17]O NMR spectroscopy would be a powerful probe[30] of $CO_2$ adsorption modes as the two oxygen atoms per $CO_2$ molecule can have significantly different local environments depending on the adsorption product formed. The investigated adsorption products have a total of four main types of oxygen environment (Fig. 2a). Ammonium carbamate chains have two oxygen environments with one of these bound to a metal ion, carbamic acid pairs have two oxygen environments corresponding to the carbonyl and hydroxyl oxygens, and the mixed adsorption mode features all four of these oxygen environments. Importantly, as [17]O is a spin 5/2 nucleus, the resulting NMR spectra will be affected not only by the familiar chemical shift, ($\delta_{iso}$), but also by the quadrupolar interaction, i.e., the interaction of the nuclear quadrupole moment with the surrounding electric field gradient, which is defined by the parameters $C_Q$ and $\eta_Q$[31]. $C_Q$ is the quadrupolar broadening defined as $C_Q = eQV_{zz}/h$ and gives the magnitude of the interaction whilst $\eta_Q$ measures the asymmetry of the interaction as $\eta_Q = (V_{yy} - V_{xx}/V_{zz})$, where $V_{xx}$, $V_{yy}$ and $V_{zz}$ are the principal components of the electric field gradient tensor, e is the electronic charge, Q is the nuclear quadrupole moment, and h is the Planck constant. By performing a [17]O MAS NMR experiment, $\delta_{iso}$, $C_Q$ and $\eta_Q$ values can be obtained, therefore providing more information than [13]C NMR.

The calculated [17]O NMR parameters (Fig. 2, Supplementary Table 3) show that, broad clusters of data points are found for the various adsorption environments. Excitingly, the OH oxygens in carbamic acid species are differentiated from the other oxygen environments by a lower [17]O chemical shift, a higher $C_Q$ and a lower $\eta_Q$ (marked with a yellow box). This differentiation is unambiguous compared to that found by [13]C NMR (Supplementary Fig. 1). For the non-protonated oxygen present in carbamic acids (grey), two distinct groupings are

seen, corresponding to oxygen in the mixed and carbamic acid pair environments, with the former having a higher shift and $C_Q$ but a lower $\eta_Q$. Interestingly, the data also show that metal-bound carbamate oxygens (blue) are differentiated from free carbamate oxygens (red) by their generally lower chemical shifts. We note that some outliers are seen amongst the carbamate oxygens (blue and red), which correspond to ammonium carbamate chain structures with different hydrogen bonding arrangements. One of these is for (R,R)-dach–Mg$_2$(dobpdc) (dach = trans-1,2-diaminocyclohexane) where additional hydrogen bonding is seen between pairs of carbamate chains[32] and another outlier is for (i-2)-Mg$_2$(dobpdc) (i-2 = n-iso-propylethylenediamine) where the proposed structure has a hydrogen bond between an amine and the oxygen next to the metal[11]. Overall the DFT calculations show that [17]O NMR should provide good differentiation between adsorption products, especially for carbamic acid species.

## Detection of different carbon capture products with [17]O NMR

Motivated by the computational results, experimental [17]O NMR spectra were acquired for a series of representative adsorbents. First, a control experiment for activated Mg$_2$(dobpdc), i.e., with no amine functionalisation, revealed a relatively sharp resonance for physisorbed $CO_2$ at 61.5 ppm (Fig. 3a)[33].

(Ee-2)–Mg$_2$(dobpdc)–CO$_2$ (ee-2 = N,N-diethylethylenediamine) was then selected as the first amine functionalised sample, as previous characterisation has confidently assigned this material to form ammonium carbamate chains[11,17]. Excitingly, the [17]O MAS NMR spectrum supports ammonium carbamate chain formation, with two oxygen environments of similar signal intensity observed (Fig. 3b). Deconvolution of these two resonances, aided by a multiple-quantum MAS (MQMAS) spectrum (Supplementary Fig. 2), gave $\delta_{iso}$, $C_Q$ and $\eta_Q$ values in good agreement with DFT-calculated parameters for ammonium carbamate chains (Table 1). The agreement for six NMR parameters gives much greater confidence in the structural assignment than previous NMR work[17]. It is important to note that the calculated DFT structures exclude the presence of water when considering the adsorption mechanism. Care has been taken to

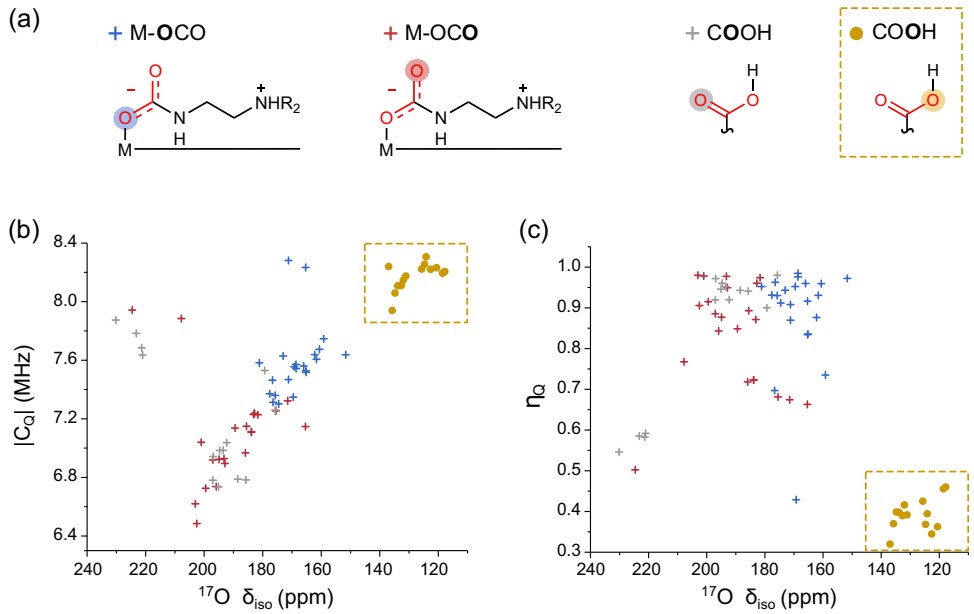

**Fig. 2 | Computation exploration of [17]O NMR as a probe of adsorption modes.** The calculated [17]O NMR parameters for the different oxygen environments present in the main adsorption structures. **a** Figures illustrating the classifications, with red and blue corresponding to the oxygens found in carbamates and grey and yellow those of carbamic acids. **b** Plot of $C_Q$ vs $\delta_{iso}$ values for the various oxygen classes. **c** Plot of $\eta_Q$ vs $\delta_{iso}$ values for the various oxygen classes. This figure shows the clear differentiation of adsorption products that is achieved with [17]O NMR. Source data are provided as a Source data file.

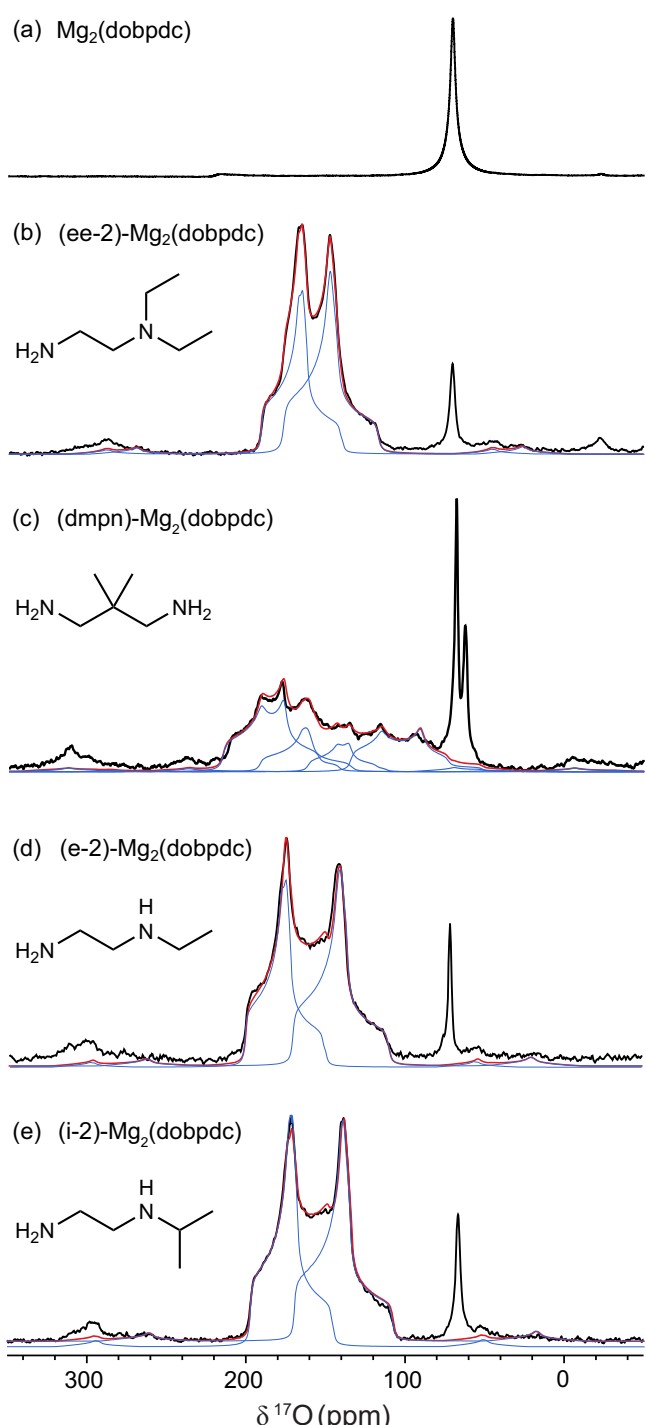

**Fig. 3 | ¹⁷O MAS NMR spectra of representative ¹⁷O-enriched CO₂-dosed adsorbents. a** The ¹⁷O MAS NMR (23.5 T, $v_0$ ¹⁷O = 135.6 MHz, 20 kHz MAS, 32768 transients, experiment time ~27 min) of activated Mg₂(dobpdc) i.e. diamine-free MOF. **b–e** The MAS ¹⁷O NMR spectra (20.0 T, $v_0$ ¹⁷O = 115.4 MHz, 14 kHz MAS, 32768 transients, experiment time ~27 min) of the four investigated diamine-functionalised frameworks. Black, red, blue lines show the experimental data, the fit, and the deconvoluted peaks, respectively. These spectra show strong support for the carbamate chain adsorption mechanism (ee-2, e-2 and i-2 analogues), for the mixed adsorption mechanism (dmpn analogue) and highlight the sensitivity of ¹⁷O NMR to different carbon capture modes.

exclude water from the system experimentally (see 'Methods'). The spectrum for (ee-2)-Mg₂(dobpdc) presented two additional peaks, one at 70.5 ppm assigned to physisorbed CO₂ and a smaller second peak at around −22 ppm. The identity of this minor peak is unknown, though it potentially arises from CO₂ reacting with defects in the metal-organic framework, since this signal was also weakly observed in the spectrum of unfunctionalised Mg₂(dobpdc). Finally, the ¹⁷O NMR spectrum could be acquired rapidly, with an acquisition time of ~27 min, and with an estimated cost of ¹⁷O enriched CO₂ gas of £50 per sample, which could be reduced in future by optimising the gas dosing line used to prepare samples. Although care has been taken to ensure adequate delays during the acquisition of these experiments, the integrated signal intensities cannot be considered truly quantitative owing to the differences in $T_1$ and $T_2$ relaxation and the different nutation rates for ¹⁷O nuclei with different quadrupolar couplings.

(Dmpn)·Mg₂(dobpdc) (dmpn = 2,2-dimethyl-1,3-diaminopropane) is hypothesised to adsorb CO₂ via a mixed adsorption mechanism (Fig. 1e)[17] and is therefore an ideal candidate to assess whether ¹⁷O NMR spectroscopy can be used to determine more complex binding modes. The ¹⁷O MAS NMR spectrum (Fig. 3c, Supplementary Figs. 2, 3) shows a broad lineshape with multiple overlapping signals, which could be deconvoluted to reveal four oxygen environments consistent with CO₂ binding via the mixed adsorption mechanism. To gain increased confidence in the extracted NMR parameters, we simultaneously fitted data from two magnetic field strengths (Fig. S4). The 12 measured NMR parameters are consistent with the DFT-calculated values for the mixed mechanism (Table 2), and provide important support for this recently hypothesised adsorption mode. Most notably, the experimental results show a clear carbamic acid OH resonance, which stands out with a lower chemical shift, larger $C_Q$ and a lower $\eta_Q$ as predicted by the DFT calculations (see Supplementary Fig. 3 for spectrum at 23.5 T, where the acid resonance is more clearly resolved). Finally, we note that for dmpn−Mg₂(dobpdc), two resonances are observed for physisorbed CO₂ at 62.7 and 68.1 ppm, with the origin of these currently unclear. Summarising, the experiments on (dmpn)−Mg₂(dobpdc) showcase the excellent ability of ¹⁷O NMR to determine complex adsorption modes and to distinguish ammonium carbamate and carbamic acid species.

CO₂-dosed (i-2)-Mg₂(dobpdc), (i-2 = N-isopropylethylenediamine), and (e-2)-Mg₂(dobpdc), (e-2 = N-ethylethylenediamine), were then investigated to further test the robustness of ¹⁷O NMR in identifying CO₂ adsorption products. The ¹⁷O MAS NMR spectra of these compounds closely resemble that of (ee-2)-Mg₂(dobpdc), consistent with ammonium carbamate chain formation (Fig. 3d, e). Interestingly, discrepancies arise when comparing the experimental and DFT-calculated NMR parameters for (e-2)-Mg₂(dobpdc) and especially (i-2)-Mg₂(dobpdc) (Table 1), suggesting that the DFT-proposed models are inaccurate. The models for (ee-2)−Mg₂(dobpdc) and (i-2)−Mg₂(dobpdc) (based on single crystal diffraction structures for the analogous zinc frameworks)[11] show important differences in their hydrogen bonding arrangements, with a hydrogen bond formed from the ammonium to the "free" carbamate oxygen for the ee-2 variant (Supplementary Fig. 5a), and the metal bound carbamate oxygen for the i-2 variant (Supplementary Fig. 5b). In the DFT model for (e-2)−Mg₂(dobpdc) additional hydrogen bonding interactions are also present. The findings that (i) the experimental ¹⁷O NMR parameters for (i-2)-Mg₂(dobpdc) and (e-2)-Mg₂(dobpdc) show poor agreement with the DFT values, and (ii) the spectra closely resemble that for (ee-2)-Mg₂(dobpdc), challenge these alternative hydrogen bonding arrangements and suggests that all structures instead have hydrogen bonds solely between the ammonium proton and the "free" oxygen as in (ee-2)−Mg₂(dobpdc) (Supplementary Fig. 5a). DFT parameters for an improved (e-2)-Mg₂(dobpdc) model are shown in Supplementary Fig. 6

**Table 1 | Overview of the experimental and DFT-calculated $^{17}O$ NMR parameters for the various diamine-functionalised frameworks adsorbing $CO_2$ to form carbamate chains**

| Compound | Amine structure | $\delta$ $^{17}O$ (ppm)<br>Experiment (DFT) | $C_Q$ (MHz)<br>Experiment (DFT) | $\eta_Q$<br>Experiment (DFT) |
|---|---|---|---|---|
| (ee-2)–$Mg_2$(dobpdc) | | M-**O**CO: 177 (169) | 6.8 (7.4) | 1.0 (1.0) |
| | | M-OC**O**: 191 (186) | 6.4 (7.0) | 0.8 (0.7) |
| (i-2)–$Mg_2$(dobpdc) | | M-**O**CO: 171 (169) | 6.9 (7.6) | 1.0 (0.4) |
| | | M-OC**O**: 197 (224) | 6.3 (7.9) | 0.9 (0.5) |
| (e-2)–$Mg_2$(dobpdc) | | M-**O**CO: 172 (174) | 6.9 (7.3) | 0.9 (0.9) |
| | | M-OC**O**: 202 (175) | 6.5 (7.3) | 0.9 (0.7) |

The DFT structures used for the different compounds correspond to those in Supplementary Fig. 5. We estimate that the experimental errors are ±1 ppm, ±0.2 MHz, and ±0.1, for $\delta$ $^{17}O$, $C_Q$, and $\eta_Q$, respectively. Bold letters are used to indicate the oxygen investigated.

**Table 2 | The experimental and DFT calculated $^{17}O$ parameters for dmpn-$Mg_2$(dobpdc)**

| Compound | Amine structure | $\delta$ $^{17}O$ (ppm)<br>Experiment (DFT) | $C_Q$ (MHz)<br>Experiment (DFT) | $\eta_Q$<br>Experiment (DFT) |
|---|---|---|---|---|
| (dmpn)–$Mg_2$(dobpdc) | | M-**O**CO: 168 (166) | 6.9 (7.6) | 0.8 (1.0) |
| | | M-OC**O**: 194 (183) | 7.2 (7.2) | 0.7 (0.9) |
| | | C**O**OH: 217 (230) | 7.6 (7.9) | 0.6 (0.5) |
| | | CO**O**H: 137 (131) | 8.0 (8.2) | 0.3 (0.4) |

DFT calculations are for the mixed carbamic acid–ammonium carbamate adsorption structure. The experimental fit was obtained by simultaneously fitting data on two independent samples at two field strengths (20.0 and 23.5 T) using ssnake software[79], see Supplementary Fig. 4. We estimate that the experimental errors are ±3 ppm, ±0.5 MHz, and ±0.2, for $\delta$ $^{17}O$, $C_Q$, and $\eta_Q$, respectively. Bold letters are used to indicate the oxygen investigated.

and Supplementary Table 4, with our findings further highlighting the excellent ability of $^{17}O$ NMR experiments to differentiate subtly different $CO_2$ adsorption products.

**Observation of carbamic acid formation in (ii-2)–$Mg_2$(dobpdc)**
As a final test of $^{17}O$ NMR as a probe of carbon capture mechanisms in MOFs, we examined the capture mode of (ii-2)-$Mg_2$(dobpdc) for the first time (ii-2 = N,N-diisopropylethylenediamine). We initially assumed this material would form ammonium carbamate chains upon $CO_2$ adsorption, as in the related material (ee-2)–$Mg_2$(dobpdc). Excitingly, the $^{17}O$ MAS NMR spectrum acquired at 23.5 T (Fig. 4a) instead reveals a clear carbamic acid resonance ($\delta_{iso}$ = 125.8 ppm, $C_Q$ = 7.99 MHz, $\eta_Q$ = 0.43, see carbamic acid OH groups in Fig. 2). This peak was also seen in a spectrum on an independent sample at 20.0 T (Supplementary Fig. 7). Furthermore, the integral of the carbamic acid peak relative to the rest of the peaks is 1:3.34, likely indicating the presence of four oxygen environments which would be expected in a mixed ammonium carbamate–carbamic acid adsorption structure. The NMR spectrum seen is different from that of dmpn–$Mg_2$(dobpdc) (Supplementary Fig. 3), hinting that this is a different mixed mechanism to that previously reported (Fig. 1e). The left-hand overlapped feature is harder to deconvolute, likely consisting of three oxygen environments at similar shifts.

Investigating this adsorption mechanism further, a $^{13}C$ MAS NMR spectrum (Fig. 4b) showed two chemisorbed $CO_2$ resonances at 163.8 and 160.1 ppm, assignable to ammonium carbamate and carbamic acid, respectively, and consistent with the observations from $^{17}O$ NMR. The $^{13}C$ peaks had relative intensities of 1:0.9 (quantitative NMR) further supporting a mixed adsorption mechanism consisting of two different $CO_2$ environments. Support for the $^{13}C$ peak assignments is provided by a 2D $^1H$–$^{13}C$ heteronuclear correlation experiment with a short contact time (Fig. 4c), which reveals $^1H$–$^{13}C$ correlations for

hydrogens nearby the carbons of the chemisorbed $CO_2$. Most importantly, the $^{13}C$ resonance at 160.1 ppm shows a strong correlation with a $^1H$ resonance at 10.3 ppm, assigned to a carbamic acid COOH group[17]. Strong N–H correlations are observed for both resonances supporting reaction with $CO_2$ at the primary amine in both cases. The 163.8 ppm $^{13}C$ resonance also shows a weak correlation with an ammonium group at 14.6 ppm, with the $^1H$ chemical shift of this species comparable to those observed previously for tertiary ammonium groups in ammonium carbamate chains for related materials[17].

Finally, an adsorption structure was proposed to explain the above results (Fig. 4d). The proposed structure features a mixture of ammonium carbamate chains and carbamic acid chains, with $CO_2$ insertion occurring between the metal-amine bond in both cases. The relative signal ratios from $^{17}O$ and $^{13}C$ NMR suggests that these two chain variants are present in similar proportions and hence have similar free energies. To check the proposed model, NMR parameters were obtained from DFT (Table 3) and showed a good agreement for the OH peak corresponding to carbamic acid, however, further work is needed to confidently assign the environments of the other resulting three peaks. Importantly, ii-2 is only the second diamine variant (after dmpn) that has been shown to form carbamic acid. In common with dmpn, ii-2 possesses bulky alkyl groups, suggesting that steric bulk is an important handle for tuning adsorption chemistry in amine-functionalised metal-organic frameworks. This may suggest that a range of diamines which contain bulky alkyl groups can access this new adsorption mechanism and it will be important to understand the kinetics of this mechanism and how it relates to the application of MOF carbon capture systems.

**Applying $^{17}O$ NMR to study $CO_2$ capture in amine-grafted silicas**
Overall, the work shown here demonstrates that $^{17}O$ NMR spectroscopy is a powerful tool for understanding $CO_2$ adsorption

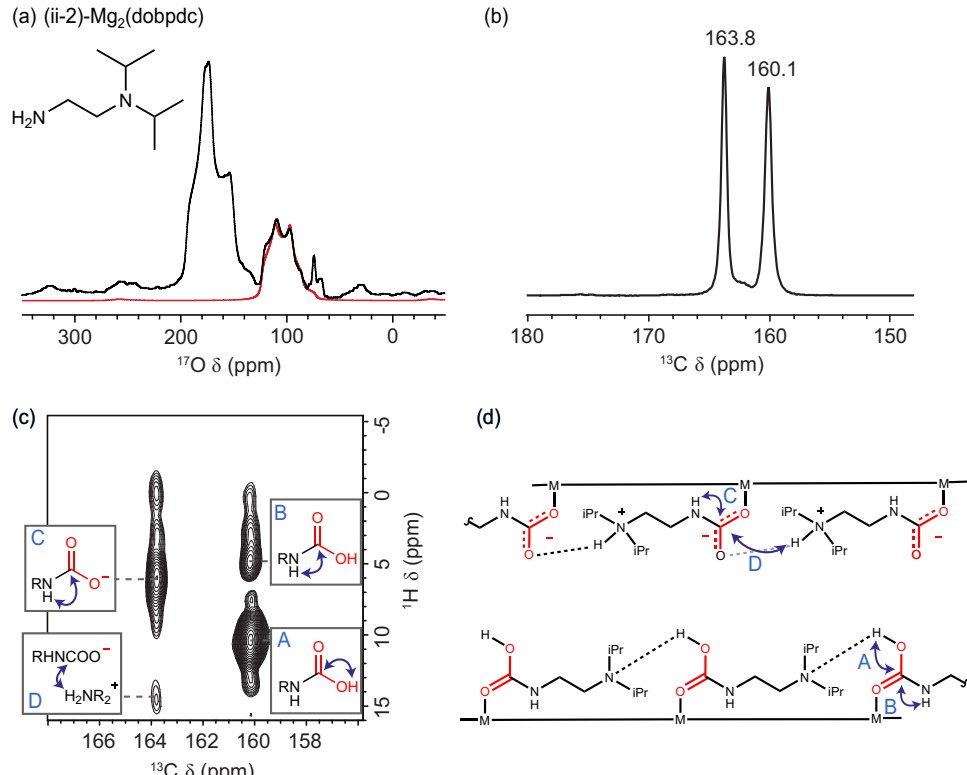

**Fig. 4 | A mixed CO$_2$ adsorption mode for (ii-2)–Mg$_2$(dobpdc). a** $^{17}$O NMR (23.5 T, $v_0$ $^{17}$O = 135.6 MHz, 20 kHz MAS, 1048576 transients, experiment time ~15 h) spectrum of (ii-2)-Mg$_2$(dobpdc) dosed with $^{17}$O-CO$_2$. The red line shows the deconvolution of a single oxygen environment assigned to a carbamic acid OH group. **b** MAS $^{13}$C NMR (16.4 T, 15 kHz MAS) spectra acquired by cross polarization (contact time 1 ms) for ii-2-Mg$_2$(dobpdc)–CO$_2$ dosed at 950 mbar $^{13}$CO$_2$. **c** $^1$H → $^{13}$C heteronuclear

correlation (contact time 100 μs) spectrum, with key correlation peaks indicated by blue arrows on hypothesised Lewis structures. **d** Lewis structure of the DFT-calculated proposed structure. Key $^1$H–$^{13}$C correlation groups are shown. These spectra show strong evidence for a previously unknown adsorption mechanism in amine-functionalised where carbamate chains and carbamic acids form on opposite chains.

---

**Table 3 | The experimental and DFT calculated $^{17}$O parameters of (ii-2)-Mg$_2$(dobpdc)**

| Compound | Amine structure | δ $^{17}$O (ppm) Experiment (DFT) | $C_Q$ (MHz) Experiment (DFT) | $\eta_Q$ Experiment (DFT) |
|---|---|---|---|---|
| (ii-2)–Mg$_2$(dobpdc) | | M-**O**CO: 185 (176) | 7.5 (7.4) | 0.9 (0.9) |
| | | M-OC**O**: 198 (203) | 7.5 (6.6) | 1.0 (1.0) |
| | | M-C**O**OH: 196 (179) | 6.8 (7.5) | 0.8 (0.9) |
| | | M-CO**O**H: 126 (137) | 8.3 (8.2) | 0.5 (0.3) |

DFT calculations are for the mixed carbamate chain structure described in Fig. 4d. The experimental fit was obtained by simultaneously fitting data on two independent samples at two field strengths (20.0 and 23.5 T) using ssnake software[79], see Supplementary Fig. 8. Bold letters are used to indicate the oxygen investigated.

---

mechanisms in MOFs. To highlight the versatility of this approach we further applied this technique to amine-grafted silica materials, another important adsorbent class for CO$_2$ separations[19,34–54]. The CO$_2$ binding modes in amine-grafted silicas remain widely debated and an array of adsorption products have been proposed, including, ammonium carbamate[34–50], carbamic acid[19,37–52], ammonium bicarbonate[34,44–51,53,54], and surface-bonded silyl carbamates[36,37,48–50].

Here, two amine-grafted silicas, propylamine-SBA15 (Pr-Si) and triamine-SBA15 (Tri-Si) (Supplementary Table 6) were synthesised and investigated by $^{17}$O MAS NMR spectroscopy at 23.4 T (Fig. 5). The $^{17}$O MAS NMR spectra show broad signals at 177.5 ppm and 28.7 ppm for Tri-Si, and 175.5 ppm and 33.3 ppm for Pr-Si, with no signal corresponding to physisorbed CO$_2$ observed in these samples. The left hand signals at ~175 ppm are consistent with ammonium carbamate oxygens, see Fig. 2 and additional DFT calculations for amine-grafted

silicas in Supplementary Table 7 and Supplementary Fig. 11, suggesting that ammonium carbamate is the major adsorption product for the two materials. The right hand resonances at ~30 ppm in the two spectra are harder to assign, however, it is clear that these signals arise from an interaction between the CO$_2$ and the amine-grafted silica material given the lack of any $^{17}$O resonances in the control experiment without CO$_2$ (Fig. 5c). We considered the previously proposed carbamic acid, ammonium bicarbonate and silylpropylcarbamate species as candidate products (Supplementary Fig. 11, Supplementary Table 7), but DFT NMR calculations gave chemical shifts that differed significantly from the experimental values, suggesting that none of these species are present in any significant quantity (Supplementary Table 7). This new resonance is therefore suggestive either of a new adsorption mode in amine-grafted silica materials, or of isotopic enrichment of oxygen atoms in the silica backbone[55,56]. Overall our experiments on

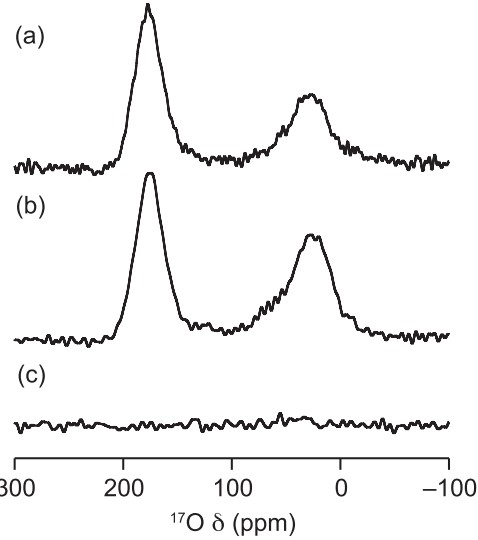

**Fig. 5 | Studying CO₂ capture by amine-functionalised silicas.** $^{17}O$ (23.5 T, $v_0$ $^{17}O$ = 135.6 MHz, 20 kHz MAS, 65536 transients, experiment time ~55 min) NMR spectrum of **a** Tri-Si and **b** Pr-Si dosed with $C_2{}^{17}O$, **c** undosed Pr-Si. These spectra show signals consistent with ammonium carbamate oxygens and show a lack of signal intensity expected for carbamic acids. Furthermore, $^{17}O$ NMR spectroscopy reveals a currently unassigned signal at $\delta_{iso}$ = 30 ppm, suggesting a previously undetermined adsorption mechanism.

amine-grafted silicas highlight the broad applicability of $^{17}O$ NMR measurements as a probe of $CO_2$ capture modes, and we envisage that these techniques will lead to the discovery of new $CO_2$ chemistry in a wide range of materials.

In conclusion, this work shows that $^{17}O$ NMR is an excellent probe of different $CO_2$ adsorption products in amine functionalised adsorbents. In particular, $^{17}O$ NMR can differentiate between ammonium carbamate chains and carbamic acids in a wide range of materials. Our measurements provide new support for ammonium carbamate chain formation in a series of (amine)–$Mg_2$(dobpdc) variants, and also provide strong evidence for a recently proposed mixed ammonium carbamate–carbamic acid mechanism for the material (dmpn)–$Mg_2$(dobpdc). We reveal carbamic acid formation in a previously poorly studied adsorbent, (ii-2)-$Mg_2$(dobpdc), highlighting the prevalence of carbamic acid in frameworks with bulky amine groups. Finally, initial measurements on amine-grafted silica materials showcase the excellent versatility of the technique, and support the formation of ammonium carbamates in these materials, while also suggesting a new adsorption mode may be in operation. It is worth noting that care was taken to ensure no water was present during the preparation of samples. It is known that the presence of water would impact the $CO_2$ adsorption mechanism and $^{17}O$ NMR spectroscopy would be a powerful tool to explore additional mechanisms further. In the future $^{17}O$ NMR spectroscopy will be extended to a range of carbon capture technologies and will ultimately enable the design of improved materials that can help tackle the climate crisis.

## Methods
### Materials
All of the chemicals used in this project were purchased from commercial suppliers and were used without further purification. The ligand 4,4′-dihydroxy-[1,1′-biphenyl]-3,3′-dicarboxylic acid (H₄dobpdc) was purchased from Hangzhou Trylead Chemical Technology. $^{17}O$-enriched $CO_2$ gas was purchased from ICON/Berry & Associates, Inc, with ~20 at.% $^{17}O$.

### Mg₂(dobpdc) synthesis
$Mg_2$(dobpdc) was synthesised according to a previously reported procedure[11]. Mg(NO₃)₂·6H₂O (11.5 g, 45.0 mmol, 1.24 equiv), H₄dobpdc (9.90 g, 36.0 mmol, 1.00 equiv), N,N-dimethylformamide (DMF) (90 mL), and methanol (110 mL) were mixed together in a 350 mL glass heavy wall pressure vessel (Chemglass, CG-1880-42). The reaction mixture was sonicated for 15 min until all of the solids had dissolved, and was then sparged with N₂ for 1 h. The reaction vessel was sealed and heated at 120 °C with stirring for 21 h. This resulted in the precipitation of a white solid from solution. The solid was collected via vacuum filtration and quickly returned to the reaction vessel along with fresh DMF (250 mL). The reaction vessel was then heated to 60 °C for 3 h with stirring. Following this, the solid was again collected via vacuum filtration and returned to the reaction vessel with fresh DMF (250 mL) and again heated to 60 °C for 3 h with stirring. This washing process with DMF was repeated a total of three times, after which the solid was washed three more times in methanol (250 mL) at 60 °C to yield the desired product, $Mg_2$(dobpdc). A small portion of the product (ca 0.1 g) was collected via filtration and activated for characterisation by powder diffraction (Supplementary Fig. 10) by heating to 60 °C in N₂ for 15 h. The remaining $Mg_2$(dobpdc) was stored in methanol.

### Diamine-functionalised Mg₂(dobpdc) synthesis
Diamine-functionalised $Mg_2$(dobpdc) materials were synthesised according to a procedure previously reported in literature[11]. Methanol-solvated $Mg_2$(dobpdc) was filtered and washed with toluene (50 mL). The filtered MOF (ca 0.1–0.4 g) was then added to a toluene (4 mL) and diamine (1 mL) solution and left to soak for at least 12 h. The solid was then collected via vacuum filtration and washed with toluene (50 mL). e-2, dmpn, ee-2, i-2 and ii-2, functionalised $Mg_2$(dobpdc) materials were activated by heating in an aluminium bead bath under N₂ to 125 °C for 1 h, 150 °C for 1 h, 125 °C for 1 h, 130 °C for 1 h, and for 130 °C for 1.5 h, respectively. This activation step removes solvent as well as excess diamine, and ensures the samples are dry prior NMR sample preparation (see below). A portion (10–20 mg) was taken for powder X-ray diffraction analysis (Supplementary Fig. 10). To determine sample stoichiometries by $^1H$ NMR (Supplementary Fig. 9, Supplementary Table 5), ~5 mg of the activated amine-functionalised MOFs was digested in a mixture of dimethyl sulfoxide (DMSO-d₆) (1 mL) and two Pasteur pipette drops of deuterated hydrochloric acid (DCl) (35 wt% in D₂O, ≥ 99 at.% D).

### Preparation of amine-grafted SBA15
12 g of Pluronic P123 triblock copolymer, 90 g of distilled water, and 360 g of 2 M HCl aqueous solution were mixed in a Teflon-lined container. The mixture was stirred at 35 °C for about 2 h, until complete dissolution of P123. Then, 25.5 g of tetraethyl orthosilicate was added to this solution under vigorous stirring. Stirring was stopped after 5 min, and the mixture was kept under static conditions at 35 °C for 20 h, followed by 48 h at 100 °C in an autoclave. The solid product was collected by filtration, washed with distilled water, dried at ambient condition, and calcined at 550 °C in flowing air for 6 h.

Amine grafting was carried out as described elsewhere[57]. The SBA-15 support was dried at 120 °C for 4 h to remove residual moisture. Then, 1.0 g of the dried support was transferred to a multineck flask, to which 30 mL of toluene was added. The mixture was stirred at ambient temperature for 2 h, and 0.3 mL of water was added dropwise. The mixing continued for an additional 2.5 h. The temperature was then raised to 110 °C, followed by addition of 1 mL of propylamine silane or 3-[2-(2-aminoethylamino)ethylamino]propyl trimethoxysilane for propylamine-SBA15 (referred to as Pr-Si) and triamine-SBA15 (referred to as Tri-Si), respectively. The mixture was kept under reflux overnight. The grafted materials were filtered and washed with toluene followed by pentane, then dried at room temperature overnight, and archived in

sealed vials. Activation before gas dosing for NMR studies was carried out by heating at 120 °C under flowing nitrogen for at least 1 h.

## C$^{17}$O$_2$ dosing of amine functionalised adsorbents

The activated amine functionalised adsorbents were packed into 4 mm or 3.2 mm NMR rotors inside a nitrogen-filled glovebag, thereby excluding water as far as possible. Each sample was then evacuated for a minimum of 10 min in a home-built gas manifold, as described previously[17]. $^{17}$O-enriched CO$_2$ gas (20 at.% $^{17}$O) was then used to dose the samples with gas at room temperature, before sealing the rotors inside the gas manifold with a mechanical plunger. (ee-2)-Mg$_2$(dobpdc) was dosed for 0.5 h with a final gas pressure of 896 mbar. The (dmpn)-Mg$_2$(dobpdc) sample for measurements at 20.0 T was dosed for 15 h with a final gas pressure of 1253 mbar, and the second independent (dmpn)-Mg$_2$(dobpdc) sample for measurements at 23.5 T was dosed with $^{17}$O-enriched CO$_2$ for 15 h with a final gas pressure of 1113 mbar. (e-2)-Mg$_2$(dobpdc) was dosed for 1 h, and the final gas pressure was 448 mbar. (i-2)-Mg$_2$(dobpdc) was dosed for 0.5 h, and the final gas pressure was 1116 mbar. The (ii-2)-Mg$_2$(dobpdc) sample for measurements at 20.0 T was dosed for 0.75 h with a final gas pressure of 1015 mbar, and the second independent (ii-2)-Mg$_2$(dobpdc) sample for measurements at 23.5 T was dosed with $^{17}$O-enriched CO$_2$ for 0.5 h with a final gas pressure of 1039 mbar. For activated Mg$_2$(dobpdc) (i.e., with no amines), activation was first carried out by heating in flowing nitrogen gas at 180 °C for 15 h. This sample was then packed in an NMR rotor (as above) and dosed with gas for 0.5 h, and the final gas pressure was 1075 mbar. For silica grafted amines, samples were dosed inside 3.2 mm NMR rotors for 0.5 h with final gas pressures for Pr-Si and Tri-Si of 1083 mBar and 993 mBar, respectively.

## NMR spectroscopy

$^{17}$O MAS and MQMAS experiments were performed using Bruker spectrometers equipped with a 20.0 T wide-bore and 23.5 T standard bore magnets, corresponding to a $^1$H Larmor frequencies, $v_0$, of 850 MHz and 1 GHz. For experiments at 20.0 T, a Bruker Avance III spectrometer was used, alongside a Bruker 4 mm low-γ HX double resonance probe, and experiments were performed with an MAS frequency, $v_R$, of 14 kHz. $^{17}$O MAS spectra were acquired using a spin-echo pulse sequence with radiofrequency field strength, $v_1$, of ~50 kHz and a recycle delay of 0.05 s. NMR parameters were optimised experimentally and therefore the spin-echo experiments cannot be considered quantitative. MQMAS experiments were acquired using a z-filter pulse sequence[58–61] with triple-quantum excitation/conversion pulses with $v_1 \approx 50$ kHz and a central-transition selective π/2 pulse at $v_1 \approx 11$ kHz. All MQMAS spectra are shown after shearing using the convention described in ref. [58]. For experiments at 23.5 T, a Bruker Avance NEO spectrometer equipped with a Bruker 3.2 mm HX double resonance probe was used with a MAS rate of 20 kHz. $^{17}$O MAS spectra were acquired using a spin-echo pulse sequence, with $v_1 = 25$ kHz, and with a recycle delay of 0.05 s. Chemical shifts are given in ppm, and are referenced relative to liquid H$_2$O at 0 ppm.

## DFT calculations

The candidate structures were first geometry optimised using CASTEP[21,62–73]. This was done with (i) plane-wave basis set with an 80 Ry (1088 eV) cut-off energy, (ii) the on-the-fly generated ultrasoft pseudopotential (C17), (iii) a 1×1×3 k-point grid, (iv) the Perdew-Burke-Ernzerhol (PBE) functional with a G06 Van de Waals correction.

The NMR parameters were calculated using the same parameters and this gave values of $\delta_{iso}$, anisotropy, asymmetry, $C_Q$ and $\eta_Q$ which converged within 0.1 ppm, 0.25 ppm, 0.001, 0.0 MHz and 0.0, respectively, for the investigated oxygen and carbon nuclei at the selected k-point grid and cutoff energy.

For $^{13}$C and $^{17}$O NMR, the principal components of the chemical shielding tensor ($\sigma_{11}$, $\sigma_{22}$ and $\sigma_{33}$ where $\sigma_{33} \geq \sigma_{22} \geq \sigma_{11}$) were obtained directly from the CASTEP calculations, in terms of $\sigma_{xx}$, $\sigma_{yy}$ and $\sigma_{zz}$ where $|\sigma_{zz} - \sigma_{iso}| \geq |\sigma_{xx} - \sigma_{iso}| \geq |\sigma_{yy} - \sigma_{iso}|$. The principal components of the chemical shielding tensor were converted to chemical shift principal components using $\delta = -(\sigma_{calc} - \sigma_{ref})$ where the reference values for $^{13}$C and $^{17}$O were 171.2 and 249.8 ppm, respectively. These values were obtained from CASTEP calculations on cocaine ($^{13}$C)[74] and the amino acids tyrosine and valine ($^{17}$O)[75], and correlation of the calculated values with the experimental values with a linear fit with a fixed gradient of –1.

## Gaussian calculations

The cluster models were created using Avogadro[76] based off the models given in ref. [23]. Dangling silicon bonds at the surface edges were terminated by OH species[50]. All calculations were performed using the Gaussian 09 software[77]. Geometry optimisations and frequency calculations were performed on the model structures prior to the calculation of NMR parameters. Note: no imaginary values were observed in the frequency calculations, and as such the structures were determined to be at the true minima. All calculations were carried out at the CAM-B3LYP/pcS-2 level of theory.

## Thermogravimetric analysis

CO$_2$ uptake measurements were carried out using a thermogravimetric analyser (Q500, TA Instruments). Samples (10–20 mg) were pretreated in flowing nitrogen at 120 °C for 2 h, to remove residual impurities. After cooling the samples to 25 °C, the purge gas was switched to 15% CO$_2$ in N$_2$. After 1 h, the temperature was raised to 50, then 75 °C. The CO$_2$ uptake at each temperature was calculated based on the corresponding weight gain. The samples were then heated to 700 °C under N$_2$ gas, before switching to air for 30 min to remove any residual carbon deposit. Amine content was determined based on the weight loss during decomposition.

## Data availability

The NMR, diffraction, and structural data generated in this study have been deposited in the Cambridge Research Repository, Apollo, under accession code https://doi.org/10.17863/CAM.83965[78]. Source data are provided with this paper.

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

## Acknowledgements

This work was also supported by a UKRI Future Leaders Fellowship to A.C.F. (MR/T043024/1, A.C.F. and S.M.P.). We thank the Yusuf Hamied Department of Chemistry at Cambridge for the award of a BP Next Generation Fellowship (A.C.F.), the NanoDTC ESPSRC Grant EP/S022953/1 (M.I.M.S and A.C.F.), the financial support of the Natural Sciences and Engineering Research Council of Canada (NSERC) (A.S.).

The UK High-Field Solid-State NMR Facility used in this research was funded by EPSRC and BBSRC (EP/T015063/1), as well as the University of Warwick including via part funding through Birmingham Science City Advanced Materials Projects 1 and 2 supported by Advantage West Midlands (AWM) and the European Regional Development Fund (ERDF), as well as, for the 1 GHz instrument, EP/R029946/1 (A.C.F.). This work was performed using resources provided by the Cambridge Service for Data Driven Discovery (CSD3) operated by the University of Cambridge Research Computing Service (www.csd3.cam.ac.uk), provided by Dell EMC and Intel using Tier-2 funding from the Engineering and Physical Sciences Research Council (capital grant EP/P020259/1), and DiRAC funding from the Science and Technology Facilities Council (www.dirac.ac.uk) (A.C.F. Additional computational resources were supported by the KIST Institutional Program (Project No. 2E31201) and KISTI Super-computing Centre (Project No. KSC-2020-CRE-0189) (J.-H.L.). We thank Halle N. Redfearn for carrying out initial $^{13}C$ NMR measurements on ii-2–$Mg_2$(dobpdc). We thank Prof. Jeffrey Reimer, Prof. Jeffrey Long and Prof. Phillip Milner for helpful discussions on amine-functionalised MOFs. For the purpose of open access, the author has applied a Creative Commons Attribution (CC BY) licence to any Author Accepted Manuscript version arising.

## Author contributions

A.H.B. and S.M.P. contributed equally to this work. A.H.B., S.M.P., Z.L., J.-H.L., C.J.P. and A.C.F. carried out DFT calculations and analysis. M.I.M.S., A.C.F. and S.M.P. carried out MOF synthesis and characterisation. C.K. and A.S. carried out amine-grafted silica synthesis and characterisation. S.M.P. carried out solid-state NMR measurements, and S.M.P., A.H.B., and A.C.F. analysed the NMR data. A.H.B. and S.M.P. wrote the manuscript with contributions from all the coauthors. A.C.F. designed the study.

## Competing interests

The authors declare no competing interests.
