## [Peer Review File · Nature Communications]

Revealing Carbon Capture Chemistry with 17-Oxygen NMR SpectroscopyREVIEWER COMMENTS

Reviewer #1 (Remarks to the Author):

In this study, the authors performed theory work on the chemical shifts of the ^{17}O moieties for various components in the CO_2 -amine chemistry, and tested it experimentally on several systems. I find the study quite important and would suggest that it would be published after some revisions. Yes, ^{17}O is expensive but still doable for well-funded labs and much less expensive than the large-scale facilities quite often used.

Comments and suggestions

>> Concerning the statement that the ^{13}C spectra are broad, I would recommend softening that proposition. For moist aminated solids the ^{13}C spectra are quite narrow if the spinning speed is sufficiently high: <https://pubs.acs.org/doi/full/10.1021/jacs.8b04520>. Also ^1H NMR (high spinning speed) is quite helpful <https://pubs.rsc.org/en/content/articlehtml/2021/ma/d0ma00658k>, which could be highlighted more in the manuscript.

>> If possible some aspects of the water could be mentioned. The water of course is critically important to this kind of chemistry when present.

>> For the generality, and that this journal has a quite wide readership I would suggest also relating this study to findings in the protein domain of science for solid state ^{17}O NMR spectroscopy. Maybe <https://pubs.rsc.org/en/content/articlehtml/2022/sc/d1sc06060k> could be of relevance, or some other paper/review etc. according to the taste/opinions of the authors.

>> Technicality: For ^{17}O MAS, the spin-echo pulse sequence would normally require a weak field (central transition selective), which was used in the MQMAS experiments. I was surprised why a medium strong field was used for the spin echo, which should have rendered the spinning side band not possible to phase. Maybe in the next study, a small w_1 -field could be used also for the spin echoes.

>> I would have used an (in my view) appropriate core-property basis set (example, from the pcS-n family) instead of the archaic B3LYP/6-311+G(d,p) one. (Nothing wrong with Sir Pople though.) For this particular journal, I expect (a detailed comment or) preferably a recalculation of the Gaussian Calculations.

Reviewer #2 (Remarks to the Author):

This work presents quadrupolar ^{17}O NMR spectroscopy of amine-functionalized MOFs being used for studies of CO_2 capture. This is an area of active interest, and the researchers have expanded efforts beyond conventional $^1\text{H}/^{13}\text{C}/^{15}\text{N}$ NMR spectroscopy to look for opportunities with ^{17}O .

This is very important work because ^{17}O is an NMR isotope that has only recently become tractable, through advances in ultra-high-field NMR. Furthermore, quadrupolar species have extra sensitivity to the electronic environment surrounding the site being interrogated, especially in sites such as those proposed – with or without hydrogen bonding.

The authors admittedly explore chain formation via ^{17}O NMR – something that Forse and co-workers have reported previously in other nuclei. What is new, in particular, is a set of assignments to carbamic acid. This is part of an ongoing debate in the literature of carbon capture involving amines, since both carbamate and carbamic acid appear in the same region for ^{13}C (for example). Consequently, the exploration of ^{17}O is a key species that can better illustrate those interactions (whether the $-\text{COO}$ group is the anionic species from an ion pair, or a hydrogen-bonded $-\text{COO}(\text{H})$).

The computational data of Figure 2 is very compelling, with the clusters of data showing convincing trends.

With all these positive aspects, the very last part of the paper on amine-functionalized silicas was less compelling and felt too speculative (given the inability to assign one of the resonances). The

experimental data (Figure 5) were not well-resolved. The authors comment that DFT NMR calculations gave chemical shifts that differed significantly – reflecting the difficulty of modeling a heterogeneous environment that often requires the amine, the surface silica, and (surface) water to be accounted for in a DFT model. Aren't there model compounds that could offer a route to assign the resonance(s)? The authors discuss many of the chemisorption candidate products, but there is a larger list of candidates found in the amine-functionalized silica literature. It may be worthwhile to consider these – including possibly physisorbed species, and chemisorption products affected by the presence of water or bicarbonates.

Reviewer #3 (Remarks to the Author):

This article presents exciting NMR measurements coupled with quantum simulations that elucidate new CO₂ adsorption pathways in MOFs and in amine-functionalized silica. These techniques are novel and show great promise for uncovering a wide range of adsorption phenomena for oxygen-bearing species (CO₂, CO, etc) in functionalized porous media. Such studies are sorely needed to fundamentally understand how, e.g., CO₂ capture media and catalysts bind and release gases or other fluids, and thus move us toward practical carbon capture and other sustainable energy technologies. The NMR data is all well collected, and it is sensibly and synergistically combined with high quality DFT simulations. While there is not ideal agreement between all simulations and NMR data and some further improvements to these techniques can be made, many conclusions about CO₂ adsorption structures/mechanisms are confidently possible with these methods. These important mechanistic conclusions are not currently accessible by other techniques. This article is also nicely written overall and the concepts are conveyed clearly. I highly recommend publication in Nature Communications after minor revisions. The authors should carefully consider and address the following points when revising.

- 1) The Fig. 2 caption is mislabeled. Please add a legend in the plots, and/or make the symbols (blue +, red +, etc) in the part (a) structures LARGER or somehow more prominent.
- 2) Please relabel the spectra in Fig. 3 so that they each have a figure letter (a, b, c, d...). This makes discussion and references to them later simpler. These can appear as labels in Table 1, for example just below the structure name as "(Figure 3b)".
- 3) For all figure captions, I strongly suggest adding a sentence at the end of each stating the overall punchline or significance of the figure. How does each figure contribute to the overall conclusions of the article? This helps to orient the reader and enable faster assimilation of concepts when reading, as well as how the data relate to those concepts.
- 4) The weak peak for the (ee-2)-Mg₂(dobpdc) and in some of the other spectra of Fig. 3b is attributed to defect sites and/or physisorbed CO₂. How do we understand the relative intensities of the peaks in this MAS experiment? I assume there is no decoupling, the T₁ values are similar and the excitation pulses cover the entire spectral bandwidth with high efficiency. If so, there should be no differential signal weighting. Since most experiments that employ MAS (combined with multiple pulses and/or CP, etc) do have differential signal weighting It would be worth mentioning that the signal strengths are linear in the concentration of each distinct nuclear species. Assuming this is the case, this is in contrast to X-ray, IR, and other techniques, and thus this NMR method can directly and fortuitously quantify things like defect density and specific site abundances.
- 5) For all NMR experiments, please state 17O Larmor frequencies and typical numbers of scans and/or total experiment times. Please also state typical ranges of T₁ values for 17O-labeled CO₂ in these samples, so that readers can assess whether recycle delays are appropriate. I see a "spin-echo sequence" was used for 17O MAS experiments, which will decrease the effective excitation bandwidth for a given pulse power, and may differentially affect signal intensities. Is this a solid echo or?? Please specify and/or include a reference.
- 6) I find it unsettling that there are no error estimates or discussion/analysis in the paper. I can understand for DFT, since errors may be hard to define, but for NMR shifts and quadrupolar parameters, they should have error estimates. Indeed, how can we assess how well NMR and DFT agree if we have no idea what are the errors in either quantity?
- 7) What happens when you heat these structures? Assumedly, a family of 17O (and 13C) NMR spectra as a function of temperature will provide very useful information that will impact thermal desorption schemes or storage at different temperatures. It seems this should be mentioned in the

paper, and/or you could explore at least one other temperature for one sample to show the potential of such a study.

8) The main body text sentence just before Table 3 is very important for the scope and outlook of this paper. Please expand this and similar statements if you can. These are real scientific punchlines with exciting implications!

Response to peer reviews

Reviewer #1:

>> Concerning the statement that the ^{13}C spectra are broad, I would recommend softening that proposition. For moist aminated solids the ^{13}C spectra are quite narrow if the spinning speed is sufficiently high: <https://pubs.acs.org/doi/full/10.1021/jacs.8b04520>. Also ^1H NMR (high spinning speed) is quite helpful <https://pubs.rsc.org/en/content/articlehtml/2021/ma/d0ma00658k>, which could be highlighted more in the manuscript.

We agree with the reviewer that the resonances in the ^{13}C NMR spectra are narrow at sufficiently high spinning speeds. We believe some confusion has arisen when describing the resolution of different signals with similar ^{13}C chemical shifts (opposed to broadening of signals owing to anisotropic effects). We have altered the text so that it will be clearer for the reader.

The phrase “with the signals from these species showing very similar ^{13}C chemical shifts” has been added.

>> If possible some aspects of the water could be mentioned. The water of course is critically important to this kind of chemistry when present.

Great care has been taken to exclude water during the preparation of experimental samples. We have added greater emphasis to this in text. We agree with the reviewer that this method is potentially an exciting method for exploring these mechanisms in the presence of water and we have added text to highlight this potential application to the conclusions.

The text “It is important to note that the calculated DFT structures exclude the presence of water when considering the adsorption mechanism. Care has been taken to exclude water from the system experimentally.” has been added to the discussion of Figure 3 and “It is worth noting that great care was taken to ensure no water was present during the preparation of samples. It is known that the presence of water would impact the CO_2 adsorption mechanism and ^{17}O NMR spectroscopy would be a powerful tool to explore additional mechanisms further.” has been added to the conclusion.

>> For the generality, and that this journal has a quite wide readership I would suggest also relating this study to findings in the protein domain of science for solid state ^{17}O NMR spectroscopy. Maybe <https://pubs.rsc.org/en/content/articlehtml/2022/sc/d1sc06060k> could be of relevance, or some other paper/review etc. according to the taste/opinions of the authors.

The reference to proteins/biological systems is somewhat outside the scope of this paper which focusses heavily on the applicability of ^{17}O NMR to carbon capture materials.

>> I would have used an (in my view) appropriate core-property basis set (example, from the pcS-n family) instead of the archaic B3LYP/6-311+G(d,p) once. (Nothing wrong with Sir Pople though.) For this particular journal, I expect (a detailed comment or) preferably a recalculation of the Gaussian Calculations.

Gaussian calculations have been recomputed using the basis set, pcS-2. Along with the reviewer, Afonso et al. (Environ. Sci. Technol. 2019, 53, 5, 2758–2767) suggest this is a more suitable basis set for calculating NMR chemical shifts. The output did not change significantly (^{17}O d_{iso} varied by ~ 10-20 ppm) and does not change the conclusions of this work.

Reviewer #2:

With all these positive aspects, the very last part of the paper on amine-functionalized silicas was less compelling and felt too speculative (given the inability to assign one of the resonances). The experimental data (Figure 5) were not well-resolved. The authors comment that DFT NMR calculations gave chemical shifts that differed significantly – reflecting the difficulty of modeling a heterogeneous environment that often requires the amine, the surface silica, and (surface) water to be accounted for in a DFT model. Aren't there model compounds that could offer a route to assign the resonance(s)? The authors discuss many of the chemisorption candidate products, but there is a larger list of candidates found in the amine-functionalized silica literature. It may be worthwhile to consider these – including possibly physisorbed species, and chemisorption products affected by the presence of water or bicarbonates.

The observation of this signal has similarly puzzled us. We have calculated the ^{17}O NMR parameters for a further 5 adsorption candidates, suggested in the review paper by Mafra et al. (Environ. Sci. Technol. 2019, 53, 5, 2758–2767). This new data has been added to Table S7 and Figure S11. Like the previously investigated candidates, these new structures do not yield ^{17}O NMR parameters consistent with the observed resonance at ~30 ppm. Review of the literature suggests that these ^{17}O d_{iso} values are consistent with alcohols, silanols or Si-O-Si bonds. The presence of ^{17}O in these species would suggest a chemical reactivity we cannot currently explain by the previously established adsorption mechanisms, therefore, given the data acquired we do not believe it is appropriate to assign this resonance and the formation of this adsorption product should be explored in future work.

We have modified the text on this point in the revised manuscript: “This new resonance is therefore suggestive either of a new adsorption mode in amine-grafted silica materials, or of isotopic enrichment of oxygen atoms in the silica backbone.^{56,57}”

Reviewer #3:

- 1) The Fig. 2 caption is mislabeled. Please add a legend in the plots, and/or make the symbols (blue +, red +, etc) in the part (a) structures LARGER or somehow more prominent.*
- 2) Please relabel the spectra in Fig. 3 so that they each have a figure letter (a, b, c, d...). This makes discussion and references to them later simpler. These can appear as labels in Table 1, for example just below the structure name as "(Figure 3b)".*

The suggested edits to Figures 2 and 3 have been made.

- 3) For all figure captions, I strongly suggest adding a sentence at the end of each stating the overall punchline or significance of the figure. How does each figure contribute to the overall conclusions of the article? This helps to orient the reader and enable faster assimilation of concepts when reading, as well as how the data relate to those concepts.*

Such statements have been added to the figure captions as below:

Figure 2: “This figure shows the clear differentiation of adsorption products that be achieved with ^{17}O NMR.”

Figure 3: “These spectra show strong support for the carbamate chain adsorption mechanism (ee-2, e-2 and i-2 analogues), for the mixed adsorption mechanism (dmpn analogue) and highlight the sensitivity of ^{17}O NMR to different carbon capture modes.”

Figure 4: “These spectra show strong evidence for a previously unknown adsorption mechanism in amine-functionalised where carbamate chains and carbamic acids form on opposite chains.”

Figure 5: “These spectra show signals consistent with ammonium carbamate oxygens and show a lack of signal intensity expected for carbamic acids. Furthermore, ^{17}O NMR spectroscopy reveals a currently unassigned signal at $d_{\text{iso}} = 30$ ppm, suggesting a previously undetermined adsorption mechanism.”

Figure 1 already had a summarising statement in the figure caption.

4) The weak peak for the (ee-2)-Mg2(dobpdc) and in some of the other spectra of Fig. 3b is attributed to defect sites and/or physisorbed CO₂. How do we understand the relative intensities of the peaks in this MAS experiment? I assume there is no decoupling, the T₁ values are similar and the excitation pulses cover the entire spectral bandwidth with high efficiency. If so, there should be no differential signal weighting. Since most experiments that employ MAS (combined with multiple pulses and/or CP, etc) do have differential signal weighting it would be worth mentioning that the signal strengths are linear in the concentration of each distinct nuclear species. Assuming this is the case, this is in contrast to X-ray, IR, and other techniques, and thus this NMR method can directly and fortuitously quantify things like defect density and specific site abundances.

These NMR spectra cannot be considered fully quantitative owing to the differences in nutation rates of nuclei with different quadrupolar couplings, loss of signal intensity due to T₂ and T₁ relaxation. Quantification could be achieved by measuring the relaxation times and retroactively calculating the contribution to the signal differences; however, this has not been completed in this instance. However, we have qualitatively tried to account for such differences by using experimentally optimised parameters, such as recycle delays. Text has been added to clarify this point for the reader.

Text added “NMR parameters were optimised experimentally and therefore the spin-echo experiments cannot be considered quantitative”.

5) For all NMR experiments, please state ^{17}O Larmor frequencies and typical numbers of scans and/or total experiment times.

The numbers of scans, experiment times, and Larmor frequencies have been added to the figure captions for all ^{17}O NMR experiments.

Please also state typical ranges of T₁ values for ^{17}O -labeled CO₂ in these samples, so that readers can assess whether recycle delays are appropriate. I see a "spin-echo sequence" was used for ^{17}O MAS experiments, which will decrease the effective excitation bandwidth for a given pulse power, and may differentially affect signal intensities. Is this a solid echo or?? Please specify and/or include a reference.

T₁ relaxation measurements were not performed on these samples. Instead, the recycle delays were experimentally optimised by varying the recycle delay and assessing the relative signal intensity. It was found for all samples 0.05 s was sufficient.

A spin echo pulse sequences was used in all cases, rather than solid echo pulse sequence, as detailed in the Methods. See also point 4 above – we do not claim that the ^{17}O NMR spectra are fully quantitative.

6) I find it unsettling that there are no error estimates or discussion/analysis in the paper. I can understand for DFT, since errors may be hard to define, but for NMR shifts and quadrupolar parameters, they should have error estimates. Indeed, how can we assess how well NMR and DFT agree if we have no idea what are the errors in either quantity?

Experimental error estimations have been added to the appropriate table captions. E.g. from Table 1: “We estimate that the experimental errors are ± 1 ppm, ± 0.2 MHz, and ± 0.1 , for $\delta^{17}\text{O}$, C_Q , and η_Q , respectively.”

7) What happens when you heat these structures? Assumedly, a family of ^{17}O (and ^{13}C) NMR spectra as a function of temperature will provide very useful information that will impact thermal desorption schemes or storage at different temperatures. It seems this should be mentioned in the paper, and/or you could explore at least one other temperature for one sample to show the potential of such a study.

While we agree with the reviewer that this method may be a good probe for thermal desorption of CO_2 , we believe this is outside of the scope of this paper, where our focus is on showing the unique power of ^{17}O solid-state NMR spectroscopy as a sensitive probe of carbon capture chemistry.

8) The main body text sentence just before Table 3 is very important for the scope and outlook of this paper. Please expand this and similar statements if you can. These are real scientific punchlines with exciting implications!

The suggested edit have been added.

“This may suggest that a range of diamines which contain bulky alkyl groups can access this new adsorption mechanism and it will be important to understand the kinetics of this mechanism and how it relates to the application of MOF carbon capture systems.” has been added.

REVIEWERS' COMMENTS

Reviewer #1 (Remarks to the Author):

I am happy with the new version of this manuscript and has no more input.

Reviewer #2 (Remarks to the Author):

I appreciated the opportunity to review the comments from other referees. Some points were raised that were relevant.

I believe the authors have satisfied my concerns and have addressed those of other referees. It think the revised language to highlight that the new resonance in figure 5 is still "unassigned" is a good practice, especially since ^{17}O is still an under-explored species.

I'm supportive of publication, and I think this work will be of broad interest and illustrative of the utility of solid-state ^{17}O NMR at high field.

Reviewer #3 (Remarks to the Author):

This is a second review of this paper. The authors have done a nice job of revising this paper based on the previous reviews, and I recommend publication in its present form.

One additional comment related to Review 1's comment about water: How has water been excluded or removed from these samples? I cannot see significant discussion of this in the Methods section. Water appears to be involved in the synthesis procedures. I would suggest that some details of glove box handling, and/or more detail on removal of water under vacuum at elevated temperature (at least 80 C for 24 h) should be considered/included. Or, if specific steps (like washing) of the synthesis procedures remove water, then please point out those steps and justify this assertion. The statements inserted after review are not specific enough to inspire confidence.

Reviewer #3 (Remarks to the Author):

This is a second review of this paper. The authors have done a nice job of revising this paper based on the previous reviews, and I recommend publication in its present form.

One additional comment related to Review 1's comment about water: How has water been excluded or removed from these samples? I cannot see significant discussion of this in the Methods section. Water appears to be involved in the synthesis procedures. I would suggest that some details of glove box handling, and/or more detail on removal of water under vacuum at elevated temperature (at least 80 °C for 24 h) should be considered/included. Or, if specific steps (like washing) of the synthesis procedures remove water, then please point out those steps and justify this assertion. The statements inserted after review are not specific enough to inspire confidence.

The removal of residual solvent (including water) from the MOF pores is detailed in the Methods section, where we use high temperature activation steps. These activation procedures are standard for these materials:

“The solid was then collected via vacuum filtration and washed with toluene (50 mL). e-2, dmpn, ee-2, i-2 and ii-2, functionalised Mg₂(dobpdc) materials were activated by heating in an aluminium bead bath under N₂ to 125 °C for 1 h, 150 °C for 1 h, 125 °C for 1 h, 130 °C for 1 h, and for 130 °C for 1.5 h, respectively.”

We have further added the below sentence for clarity here:

From the revised text: ***“This activation step removes solvent as well as excess diamine, and ensures the samples are dry prior NMR sample preparation (see below).”***

Later in the methods, the NMR sample preparation is mentioned. We have added the phrase “thereby excluding water as far as possible” for further clarification:

*“The activated amine functionalised adsorbents were packed into 4 mm or 3.2 mm NMR rotors inside a nitrogen-filled glovebag, **thereby excluding water as far as possible**. Each sample was then evacuated for a minimum of 10 min in a home-built gas manifold, as described previously.”*

Finally, we now make sure the Methods section are referred to in the main text when water is mentioned: *“Care has been taken to exclude water from the system experimentally (see Methods).”*